# Fucosterol Isolated from Dietary Brown Alga *Sargassum horneri* Protects TNF-α/IFN-γ-Stimulated Human Dermal Fibroblasts Via Regulating Nrf2/HO-1 and NF-κB/MAPK Pathways

**DOI:** 10.3390/antiox11081429

**Published:** 2022-07-23

**Authors:** Kirinde Gedara Isuru Sandanuwan Kirindage, Arachchige Maheshika Kumari Jayasinghe, Eui-Jeong Han, Youngheun Jee, Hyun-Jin Kim, Sun Gil Do, Ilekuttige Priyan Shanura Fernando, Ginnae Ahn

**Affiliations:** 1Department of Food Technology and Nutrition, Chonnam National University, Yeosu 59626, Korea; 218388@jnu.ac.kr (K.G.I.S.K.); 218385@jnu.ac.kr (A.M.K.J.); iosu5772@jnu.ac.kr (E.-J.H.); 2Department of Veterinary Medicine and Veterinary Medical Research Institute, Jeju National University, Jeju 63243, Korea; yhjee@jejunu.ac.kr; 3Research and Development Center, Naturetch Co., Ltd., Cheonnam-si 31257, Korea; hjmari@naturetech.co.kr (H.-J.K.); sgildo@naturetech.co.kr (S.G.D.); 4Department of Marine Bio-Food Sciences, Chonnam National University, Yeosu 59626, Korea

**Keywords:** *Sargassum horneri*, fucosterol, human dermal fibroblasts, Nrf2/HO-1, MAPK, NF-κB

## Abstract

*Sargassum horneri* is a well-known edible brown alga that is widely abundant in the sea near China, Korea, and Japan and has a wide range of bioactive compounds. Fucosterol (FST), which is a renowned secondary metabolite in brown algae, was extracted from *S. horneri* to 70% ethanol, isolated via high-performance liquid chromatography (HPLC), followed by the immiscible liquid-liquid separation, and its structure was confirmed by NMR spectroscopy. The present study was undertaken to investigate the effects of FST against oxidative stress, inflammation, and its mechanism of action in tumor necrosis factor (TNF)-α/interferon (IFN)-γ-stimulated human dermal fibroblast (HDF). FST was biocompatible with HDF cells up to the 120 μM dosage. TNF-α/IFN-γ stimulation significantly decreased HDF viability by notably increasing reactive oxygen species (ROS) production. FST dose-dependently decreased the intracellular ROS production in HDFs. Western blot analysis confirmed a significant increment of nuclear factor erythroid 2-related factor 2 (Nrf2)/ heme oxygenase-1 (HO-1) involvement in FST-treated HDF cells. In addition, the downregulation of inflammatory mediators, molecules related to connective tissue degradation, and tissue inhibitors of metalloproteinases were identified. TNF-α/IFN-γ stimulation in HDF cells increased the phosphorylation of nuclear factor-κB (NF-κB) and mitogen-activated protein kinase (MAPK) mediators, and its phosphorylation was reduced with the treatment of FST in a dose-dependent manner. Results obtained from western blot analysis of the NF-κB nuclear translocation were supported by immunocytochemistry results. Collectively, the outcomes suggested that FST significantly upregulates the Nrf2/HO-1 signaling and regulates NF-κB/MAPK signaling pathways to minimize the inflammatory responses in TNF-α/IFN-γ-stimulated HDF cells.

## 1. Introduction

Since ancient times, humans have used certain species of marine algae as culinary ingredients. *Sargassum horneri* is a type of edible marine brown algae that is abundant in the sea nearby China, Korea, and Japan. Apart from its dietary uses, *S. horneri* has gained increased attention towards research on antioxidants, and anti-inflammatory agents due to its high content of bioactive compounds, such as sulfated polysaccharides, sargachromenol, phlorotannins, phenolic substances, and proteoglycans [1,2,3]. Although *S. horneri* was first reported as a naturally available source in 1962 in the Manual of Chinese Economic Seaweeds, it is more suitable in modern medicine due to its promising protective role against bacteria, anti-proliferative effects on cancer cells, and anti-inflammatory and antioxidant activity. The huge boom of future drug developments focuses more on synergetic pharmacological effects rather than the “one-target, one-drug” approach [4]. Several studies indicated that lipid compounds, such as carotenoids and sterols, are equally beneficial to human health while these substances are prevalent in diverse brown seaweeds [5]. Prior investigation revealed the presence of fucosterol (FST) as a principal sterol in brown seaweeds [6,7]. Moreover, FST is relatively abundant in *Sargassum* spp. [8]. In particular, the protective effect of FST against acute liver injury [9], urban particulate matter-induced injury and inflammation in human lung epithelial cells [10], and particulate matter-induced skin lesions [11], as well as antioxidative and anti-inflammatory effects on macrophages [12] have been reported. However, there is no evidence that FST has an anti-inflammatory effect on human dermal fibroblasts (HDF) stimulated by tumor necrosis factor (TNF)-α/interferon (IFN)-γ.

Inflammation is a complex biological response that takes place in the body for maintaining homeostasis alongside internal or external stimuli. Under specific circumstances, inflammation can increase the risk of numerous diseases reaching a chronic level. Inflammatory responses in the skin are accompanied by the occurrence of oxidative stress from intracellular reactive oxygen species (ROS) that can damage tissues and cause chronic inflammatory illnesses [13]. TNF-α and IFN-γ are inflammatory mediators that have the ability to promote oxidative stress and inflammatory reactions in cells. It causes abnormal expression of inflammatory cytokines, chemokines, and inflammatory mediated signaling pathways [14,15]. The nuclear transcription factor kappa B (NF-κB) pathway is activated by dysregulated ROS generation in cells, which promotes phosphorylation of IκBα and NF-κB and enables the NF-κB p65 subunit to enter the nucleus from the cytosol. Moreover, the mitogen-activated protein kinase (MAPK) pathway is activated by the phosphorylation of ERK, JNK, and p38. Activation of the NF-κB and MAPK pathways collectively stimulates inflammatory responses by producing inflammatory cytokines and chemokines. The nuclear factor erythroid 2-related factor 2 (Nrf2)/heme oxygenase-1 (HO-1) signaling is considered a key pathway, which protects cells by boosting the production of antioxidant enzyme genes which is essential for defending cells from inflammatory responses [13,16,17]. Keratinocytes are the outermost live cells of the skin that are directly exposed to external stimuli which may occur inflammatory reactions and release inflammatory mediators [13]. It can affect underneath cells such as HDFs to cause inflammatory reactions. In addition, HDF cells themselves can indicate inflammation, and the production of matrix metalloproteinases (MMPs), which are involved in the degradation of connective tissue components in the skin by increasing oxidative stress [18]. Moreover, external stimuli can increase oxidative stress in HDFs, activate protective as well as apoptotic pathways, and ultimately cause abnormal inflammatory reactions.

Skincare and related nutricosmetic products which contain biologically active natural ingredients are seen as the next big innovation in the cosmetic industry [19]. The present study aims to isolate bioactive compounds by using the immiscible liquid–liquid separation of 70% ethanol extract of *S. horneri.* High-performance liquid chromatography (HPLC) was implicated to identify prominent UV absorbing peaks assuming the hexane fraction contains a significant number of sterols. The collected fraction from HPLC was identified as FST by subjecting nuclear magnetic resonance (NMR) spectroscopy. Further experiments were designed, hypothesizing that FST isolated from 70% ethanol extract of *S. horneri* possesses potent antioxidant and anti-inflammatory effects on TNF-α/IFN-γ-stimulated HDF cells by regulating the Nrf2/ HO-1 signaling pathway, thereby regulating the NF-κB, MAPK signaling, and production of inflammatory cytokines and chemokines.

## 2. Materials and Methods

### 2.1. Materials

*S. horneri* samples were collected from the western shores of Jeju Island, South Korea. Recombinant TNF-α and IFN-γ were purchased from R&D Systems (Minneapolis, MN, USA). Dulbecco’s modified eagle medium (DMEM), and a mixture of streptomycin and penicillin (P/S) as antibiotics were purchased from GibcoBRL (Grand Island, NY, USA), and Fetal bovine serum (FBS) was purchased from Welgene (Gyeongsangbuk-do, South Korea). 2,2-azino-bis (3-ethylbenzothiazoline-6-sulfonic acid) diammonium salt (ABTS), 2,2-diphenyl-1-picrylhydrazyl (DPPH), 2′7′-dichlorodihydrofluorescein diacetate (DCFH-DA), 3-(4,5-dimethylthiazol-2-yl)-2,5-diphenyltetrazolium bromide (MTT), Dimethyl sulfoxide (DMSO), bovine serum albumin (BSA), Folin and Ciocalteu’s phenol reagent, ethidium bromide, and agarose were bought from Sigma-Aldrich (St. Louis, MO, USA). D-glucose was purchased from Junsei Chemical Co., Ltd. (Tokyo, Japan). BCA protein assay kit, NE-PER^®^ nuclear and cytoplasmic extraction kit, 1-Step transfer buffer, Pierce™ RIPA buffer, protein ladder, and SuperSignal™ West Femto Maximum Sensitivity Substrate were purchased from Thermo Fisher Scientific (Rockford, IL, USA). Antibodies needed for the western blot analysis were purchased from Santa Cruz Biotechnology Inc. (Dallas, TX, USA) and Cell Signaling Technology Inc. (Beverly, MA, USA). Skim milk powder was obtained from BD Difco™ (Sparks, MD, USA). Normal goat serum, Prolong^®^ Gold antifade reagent with DAPI reagent, and DyLihgtTM 554 Phalloidin were purchased from Cell Signaling Technology (Danvers, MA, USA). The primers for the reverse transcription-polymerase chain reaction (RT-PCR) were purchased from Bioneer Co. (Deadeock-gu, Daejeon, Korea). The remaining chemicals and reagents used were of analytical grade.

### 2.2. Sample Collection and Extraction

Samples were collected from the western shores of Jeju Island, South Korea. *S. horneri* was identified by the Biodiversity Research Institute in Jeju, South Korea (voucher specimen (SH2017J005), and was kept in the laboratory of Marine Bioresource Technology at Jeju National University, South Korea). Sands and impurities were removed by washing with running water, air-dried at room temperature, and pulverized into powder by using IKA MF10 laboratory pulverizer (Staufen, Germany). The powder was extracted into 70% ethanol for 12 h and repeated three times. After centrifugation and filtration, the crude extract was obtained by concentrating the extract in a rotary evaporator.

### 2.3. Compositional Analysis of Crude Extract

The contents of total polyphenol, total protein, and carbohydrate were determined according to the method mentioned in one of the previous studies [20]. In brief, the dried crude extract was dissolved and incubated in the dark after mixing with Folin–Ciocalteu regent. Total polyphenolic compounds were measured by comparing the absorbance values with the gallic acid standard. Total protein content was measured by using the Lowry method. For that, the dissolved sample was incubated in the dark after mixing with Folin–Ciocalteu regent. The total protein content was calculated by using bovine serum albumin as the standard. Carbohydrate content was measured by using phenol–sulphuric method. For that sample, it was dissolved and then mixed with phenol and sulphuric acid. Then, absorbance values were obtained after incubating in the dark. Carbohydrate content was calculated by comparing it with the standard glucose series.

### 2.4. High-Performance Liquid Chromatography (HPLC) Analysis and Structural Identification of Compounds

The crude extract was dissolved in water and partitioned with hexane by thoroughly mixing and equilibrating in a separatory funnel. The resulted hexane fraction was dried by removing the solvents using a rotary evaporator and resolved by HPLC system equipped with an ‘Ultimate 3000’-variable wavelength detector. Among the four different fractions, the concentrated hexane fraction was resolved in a BioBasic SEC-60 (300 × 7.3 mm) PREP column with the use of gradient acetonitrile: water (49:1) solvent system. Proton (^1^H) NMR data were used to identify and elucidate the structure of the isolated compound. For that, the compound was dissolved in CDCl_3_ and analyzed by an AVANCE III HD 400 spectrometer (400 MHz) (Bruker, Mundelein, IL, USA).

### 2.5. Cell Culture

HDF (KCLB, Seoul, Korea) were cultured and maintained in DMEM media supplemented with 25% F-12, 10% FBS, and 1% penicillin/streptomycin mixture at 37 °C with 5% of CO_2_ in a humidified atmosphere. Cells were sub-cultured once every 5 days while cell culture media was replaced once every 2 days with fresh media. When cells reached exponential growth during the passages of 3–6, they were accordingly seeded in multi-well plates, chamber slides, or cell culture dishes for further experiments.

### 2.6. Cell Viability and ROS Production Analysis

HDF cells seeded in a 96-well plate were treated with a series of FST concentrations and incubated for 1 h. Then, 10 μL of TNF-α/IFN-γ mixture in 1:1 ratio was added and incubated at 37 °C for 24 h. Then, MTT assay was conducted to investigate the cell viability as described in one of our previous publications [13]. In brief, the absorbance of formazan crystals dissolved in DMSO was measured by using a SpectraMax M2 microplate reader (Molecular Devices, Sunnyvale, CA, USA) at 570 nm. The effect of FST on intracellular ROS levels in HDF cells stimulated with TNF-α/IFN-γ was measured by 2′,7′-dichlorofluorescein diacetate (DCF-DA) assay, as described in one of the previous publications [20].

### 2.7. Western Blot Analysis

HDF cells were seeded at 2 × 10^5^ cells/mL of cell counts in 10 cm culture dishes for 24 h. Cells were stimulated with 10 ng/mL TNF-α/IFN-γ after being treated with a series of FST concentrations for 2 h. Then, cells were harvested and lysed using the NE-PER^®^ nuclear and cytoplasmic extraction kit (Thermo Scientific, Rockford, IL, USA). A BCA protein assay kit (Thermo Scientific, Rockford, IL, USA) was used to estimate the protein concentrations in cell lysate, and 30 μg of protein of each lysate were subjected to electrophoresis on 10% polyacrylamide gels. Then, we followed the same procedure as described in our previous study [20].

### 2.8. Immunofluorescence Analysis

Immunostaining was conducted according to the method described in the previous study with slight modifications [21]. In brief, a density of 1 × 10^4^ cells/chamber was used for cell culture in chamber slides, and samples were treated accordingly after 24 h of incubation in a humidified atmosphere. Then, wells were rinsed with PBS after 30 min of TNF-α/IFN-γ exposure, fixed, and incubated in blocking buffer for 1 h before being incubated overnight with primary antibodies (anti-NF-κB p65). The cells were then rinsed in PBS, and treated with Alexa Fluor^®^ 488 conjugated Anti-Mouse IgG for 2 h. Slides were covered with coverslips with Prolong^®^ Gold antifade reagent containing DAPI, after being washed in PBS. Then an EVOS M5000 (Thermo Fisher Scientific, Waltham, MA, USA) Imaging microscope was used to visualize the cells.

### 2.9. Reverse Transcription-Polymerase Chain Reaction (RT-PCR)

RT-PCR was carried out to investigate the expression of mRNA of inflammatory cytokines. The experiment was conducted following the method described by Jayasinghe et al. [13]. In brief, HDF cells were seeded in 6 cm culture dishes and incubated for 24 h at 37 °C in a humidified atmosphere. Then cells were treated with a series of concentrations of FST prior to the stimulation of TNF-α/IFN-γ. RNA was collected from the harvested cells and cDNAs were synthesized from 17.5 μL of RNA solution (2 μg/μL) in each by using TaKaRa PCR Thermal Cycler (TaKaRa Bio Inc., Otsu, Japan). PCR for corresponding cytokines was carried out for 35 cycles in a TaKaRa PCR Thermal Cycler. Then, the relevant bands of ethidium bromide-stained PCR products were visualized after agarose gel electrophoresis by using Wisd WUV-L20 UV transilluminator (Daihan Scientific Co., Gang-won-Do, Korea). The final analysis was conducted using NIH Image J software (Version No. 1, US National Institutes of Health, Bethesda, MD, USA).

### 2.10. Statistical Analysis

All statistical analyses of the study were performed using the SPSS software (Version 24.0, Chicago, IL, USA). One-way analysis of variance (ANOVA) followed by Duncan’s multiple range tests was used to evaluate the significant variations among data sets, and data were presented as the mean ± standard error of the mean (SEM). In this study, *p* < 0.05 was considered statistically significant.

## 3. Results

### 3.1. Extraction of S. horneri, Isolation of Fucosterol by HPLC, and Structural Elucidation

Dried *S. horneri* was ground to a powder and extracted to 70% ethanol for 24 h. The extraction yield of the *S. horneri* 70% ethanol extract (SHE) was 8.12 ± 0.26% from the initial dry weight. Of that, 3.92 ± 0.19% were carbohydrates and 1.09 ± 0.07% were proteins; meanwhile, polyphenols indicated the highest among all measured compositions, which were 14.82 ± 0.68% (Table 1). The results of this analysis do not differ significantly from previous research findings [22].

Based on the results of preliminary assessments, crude ethanol extract of *S. horneri* was further separated by the immiscible liquid–liquid separation method indicated in Figure 1A. Among all resulted fractions, the subsequent hexane fraction was subjected to HPLC based on the potent bioactivities identified through bioactivity evaluation. Seven HPLC fractions were obtained from the hexane fraction following the chromatogram presented in Figure 1B. The prominent fractions were initially screened for potent antioxidant and anti-inflammatory activities. The seventh fraction was recognized showing prominent bioactivities. HPLC was used to further purify a single peak in the chromatogram from the seventh fraction (Figure 1C). The analysis of ^1^H NMR, along with a comparison with the literature data, allowed its structure to be elucidated as FST [12]. Further, numerical values of spectral data of the ^1^H NMR, which are presented in Table 2, confirm the structure of the substance was FST. 

### 3.2. Effect of Fucosterol on Cell Viability and Intracellular ROS Production

Significant cytotoxicity of FST on HDFs was not observed up to the concentration of 120 μM (Figure 2A). As shown in Figure 2B, TNF-α/IFN-γ-stimulation significantly decreased the cell viability of HDFs. Treatment of FST on stimulated HDFs significantly and dose-dependently increased the cell viability up to 120 μM. Henceforward, 30, 60, and 120 μM concentrations of FST were used throughout the study. TNF-α/IFN-γ stimulation increased intracellular ROS in HDFs, while treatment of FST significantly and dose-dependently reduced the ROS production (Figure 2C). This result was strengthened by the findings of DCF-DA flow cytometric analysis and DCF-DA fluorescence imaging which are illustrated in Figure 2D,E, respectively. FACs analysis with the use of fluoroprobe is a reliable method of cell sorting due to its ability to omit the error caused by cell death. Rightward shifting of the peaks on FITC-A axis is reduced with the increase in FST concentrations. Meanwhile, increased green fluorescence with TNF-α/IFN-γ-stimulation compared to the control cells was gradually reduced in the FST-pre-treated HDF cells. DCF-DA flow cytometric analysis, as well as fluorescence microscopy imaging, indicated the dose-dependent effect of FST on intracellular ROS production in TNF-α/IFN-γ-stimulated HDF cells. The bioactivity of FST concentrations (30, 60, and 120 μM) was compared with the positive control Indomethacin (50 μM).

### 3.3. Fucosterol Regulated the Nrf2/HO-1 Signaling

Activation of the Nrf2/ HO-1 signaling is a key pathway in the reduction of intracellular ROS generation and contributes to the regulation of the inflammatory responses and apoptosis in cells [23]. Result of the western blot analysis of Nrf2, HO-1, and NQO1, the pretreatment of FST boosted the nuclear translocation of Nrf2 in TNF-α/IFN-γ-stimulated HDFs and increased the levels of cytosolic HO-1 and NQO1 in a dose-dependent manner (Figure 3).

### 3.4. Fucosterol Downregulated Inflammatory Mediators, MMP and Tissue Inhibitors of Metalloproteinases (TIMP)

The effect of FST on the expression of inflammatory cytokines and chemokines was evaluated by using RT-PCR by examining the key inflammatory mediators. TNF-α/IFN-γ stimulation upregulated the mRNA expression levels of key inflammatory mediators compared to the control group and along with that, FST treatments downregulated the mRNA expressions of inflammatory cytokines (IL-6, IL-8, IL-13, IL-33, IL-1β, TNF-α, and IFN-γ) in HDF cells which were upregulated by the TNF-α/IFN-γ stimulation (Figure 4A,C).

Meanwhile, the expression of mRNA related to connective tissue degradation (MMP1, MMP2, MMP3, MMP8, MMP9, and MMP13) was evaluated by using RT-PCR. As denoted in Figure 4B,D, expression of the above-mentioned molecules was upregulated by the TNF-α/IFN-γ stimulation in HDF cells compared to control, and dose-dependently downregulated by the treatment of FST. TIMPs are vital for the regulation of MMP activities in fibroblasts, and the physiological functions of ECM are significantly reliant on the balance of TIMPs and MMPs [18,24]. TNF-α/IFN-γ stimulation increased the expression of TIMP1 and TIMP2 in HDF cells compared to the control, and particularly downregulated TIMP expression in a dose-dependent manner.

### 3.5. Fucosterol Regulated the NF-κB and MAPK Signaling

Activated canonical NF-κB signaling in cells are responding to the inflammation, immune response, cell proliferation, and differentiation stimulated by external stimuli [25]. Moreover, it is a well-known fact that NF-κB and MAPK are important upstream pathways that bear a major role in inflammatory responses [11]. As illustrated in Figure 5A, western blot analysis revealed that TNF-α/IFN-γ stimulation in HDF cells increase the phosphorylation of NF-κB mediators; cytosolic IκBα, NF-κB p65, and its phosphorylation is reduced with the treatment of FST in a dose-dependent manner. Besides, western blot analysis indicated that nuclear translocation of NF-κB p65 is reduced in the same manner which increased with the TNF-α/IFN-γ stimulation. Similarly, immunofluorescence analysis indicated that the FST dose-dependently reduced NF-κB p65 nuclear translocation in TNF-α/IFN-γ-stimulated HDF cells. The intense green fluorescence in immunostaining implied an increased NF-κB p65 nuclear translocation in stimulated cells (Figure 5B). In a similar manner, phosphorylated p38, ERK, and JNK expressions in TNF-α/IFN-γ stimulated HDF cells were dose-dependently downregulated with the FST (Figure 5C).

## 4. Discussion

During the last few decades, inflammation and related complications are a focus of extensive research. It has emerged as one of the hottest topics in medical science due to its impact on the human body at chronic levels, as well as the swift growth of skin health-related concerns in the public. The inflammatory response, as a whole, is considered a primary protective mechanism that takes place as a consequence of alterations in tissue homeostasis caused by a range of stimuli, such as pathogens, tissue damage, or pollutants, and includes the activation of innate and adaptive immunity in the human body [23,26]. Nonetheless, when inflammatory reactions go beyond the controllable limit and become recurring in the body, they can cause chronic inflammatory diseases, such as atopic dermatitis, cardiovascular diseases, cancer, bronchitis, and asthma, and it can be life-threatening [13]. Inflammatory and phagocytic cells produce a variety of chemical mediators and signaling molecules, such as histamine, serotonin, leukotrienes, and prostaglandins, which contribute to the onset of inflammation [27]. Scientific evidence revealed that exacerbation of inflammatory skin diseases has occurred with the excessive levels of ROS accumulation in the cellular environment. It leads to oxidative modifications and biomolecular damage, while triggering the inflammatory signaling cascades [28]. Implementing naturally available bioactive metabolites to treat numerous diseases, including skin inflammation-related disorders, has gained attention in the current context of pharmaceutical developments [10,29]. 

Brown seaweeds, such as *S. horneri*, are well-known sources of sterols, and many studies highlighted that investigating the isolation of bioactive compounds and their therapeutic potential is crucial [30,31]. Previous studies have revealed that FST isolated from edible brown alga, such as *Eisenia bicyclia* and *Undaria pinnatifida*, have potent anti-inflammatory effects on RAW 264.7 macrophages, which stimulate to produce nitric oxide inside cells by lipopolysaccharide [16,32]. FST isolated from 70% ethanol extract of *S. horneri* was used in the present study. Immiscible liquid–liquid separation of crude ethanol extract was implemented to separate compounds in the crude extract by using their solubilities in hexane, chloroform, ethyl acetate, and water. The most effective fraction was non-cytotoxic, and had a good effect of inhibition of ROS production in TNF-α/IFN-γ-stimulated HDF cells, was continued for further investigations. According to that, the hexane fraction was chosen and the most prominent peak on the HPLC chromatograph was collected. The collected fraction was dried by evaporating the solvent and dry powder was identified as FST by using ^1^H NMR analysis.

Then, a range of experiments was carried out to investigate the anti-inflammatory effect of FST isolated from *S. horneri* in TNF-α/IFN-γ-stimulated HDF cells. Many studies revealed that TNF-α/IFN-γ triggers ROS and pro-inflammatory cytokine production in vitro [13,14,33]. Dysregulated ROS production in cells leads to inflammatory gene expression by mediating redox-based activation of the NF-κB signaling pathway [16]. As indicated in the Figure 2, DCF-DA fluorometry, fluorescence microscopic imaging, as well as FACs analysis, confirmed that TNF-α/IFN-γ stimulation promptly augmented the intracellular ROS production in HDF cells. In particular, fluorometry represents the level of ROS in cells quantitatively, and flow cytometry analysis of DCF-DA is considered a trustworthy approach, whereas it measures the fluorescence intensity of each cell [18]. Different concentrations of FST (30, 60, and 120 μM) decreased the ROS levels in TNF-α/IFN-γ-stimulated HDF cells in a dose-dependent manner. 

Accumulated shreds of evidence suggest that the significance of the Nrf2/HO-1 signaling is stimuli-specific and cell type-specific [17]. Nrf2 is a redox-sensitive transcription factor that resides in the cytoplasm as an inactive complex with Kelch-like ECH-associated protein 1 (Keap1). It can regulate the production of numerous antioxidant enzymes, including HO-1, and enters into the nucleus and binds to the antioxidant response element site. Nrf2 protects diverse cells from oxidative stress by boosting the production of antioxidant enzyme genes and proteins when stimuli activate the associated pathways [34]. FST enhanced nuclear translocation of Nrf2 in a dose-dependent manner in stimulated HDFs, as expected. At the same time, the levels of HO-1 and NQO1 change in the same way. Moreover, several studies have revealed that HO-1 and its metabolites have anti-inflammatory actions that are mediated via Nrf2 [35]. According to the best understanding over the years, numerous inflammatory cytokines are overproduced when NF-κB is activated by oxidative stress. In the meantime, the elevation of HO-1 expression, which is mediated by activated Nrf2, leads to the inhibition of NF-κB signaling [23].

One of the adaptive immune responses to inflammatory stimuli is the release of pro-inflammatory cytokines and chemokines [22]. Inflammatory mediators in HDFs, in particular, play a critical role in modulating the structural integrity of the extracellular matrix of the skin by regulating MMP, collagenase, and elastase transcription [36]. A recent study conducted by Jayasinghe et al. contained evidence of the exposure of the epidermal cells to TNF-α/IFN-γ stimulating abnormal expression of cytokines, chemokines, and inflammatory mediated signaling pathways [13]. Expression levels of IL-6, IL-8, IL-13, IL-33, IL-1β, TNF-α, and IFN-γ are significantly increased in TNF-α/IFN-γ-stimulated HDF cells, while dose-dependently downregulated by the FST treatment. Aside from that, the expression of chosen MMPs was investigated further to determine the influence of FST on skin inflammation. Even though MMPs are involved in the regeneration of normal tissues, an aberrant increase in expressions of some MMPs, such as MMP1, MMP8, and MMP13, in dermal fibroblasts is associated with type I and type III collagen (ECM) degeneration [37]. Numerous studies linked IL-6 and TNF-α as implicated in the regulation of TIMP1, MMP1, and MMP9s’ mRNA expression. Furthermore, it regulates activator protein-1 (AP-1) activation, NF-κB mediators, and the expression of MMP1, MMP3, and MMP9 in HDFs [24,38,39]. RT-PCR results revealed that increased levels of MMP1, MMP2, MMP3, MMP8, MMP9, MMP13, TIMP1, and TIMP2 due to TNF-α/IFN-γ stimulation in HDFs are down-regulated with the FST treatments. Results imply that TNF-α/IFN-γ stimulation in HDFs has a higher probability of inducing skin inflammation in underlying mechanisms, while FST treatment is noteworthy in downregulating.

Furthermore, TNF-α/IFN-γ stimulation of HDFs significantly activated crucial intracellular upstream signaling molecules, such as NF-κB and MAPK. As reported, MAPK proteins are also important for ECM breakdown [18]. According to the findings from western blot analysis, the expression levels of cytosolic p-IκBα, p-NF-κB p65, and nucleic NF-κB p65 were downregulated along with the FST doses in stimulated HDFs. Results conclude that FST positively regulates the nuclear translocation of NF-κB p65, which is strengthened by the findings of immunocytochemistry analysis of NF-κB p65 nuclear translocation. TNF-α/IFN-γ stimulation of HDFs significantly increases the phosphorylation of MAPK mediators which are JNK, ERK, and p38, while FST lowers phosphorylation in a dose-dependent manner. Thus, the findings of this study reveal that FST effectively regulates the Nrf2/HO-1 signaling, NF-κB, and MAPK pathways, and, hence, the regulation of inflammatory cytokines in TNF-α/IFN-γ-stimulated HDFs.

## 5. Conclusions

Based on the present study, hexane fraction collected from the 70% ethanol extract of *S. horneri* in an immiscible liquid–liquid separation system contains FST, which shows a prominent peak at 220 nm of UV absorbance in HPLC analysis. FST isolated from the HPLC possesses potent anti-inflammatory activities on HDFs that are stimulated with TNF-α/IFN-γ in vitro. Results show that the regulation of Nrf2/HO-1, as well as NF-κB and MAPK signaling collectively, contribute to the anti-inflammatory activity of FST in stimulated HDFs.

## Figures and Tables

**Figure 1 antioxidants-11-01429-f001:**
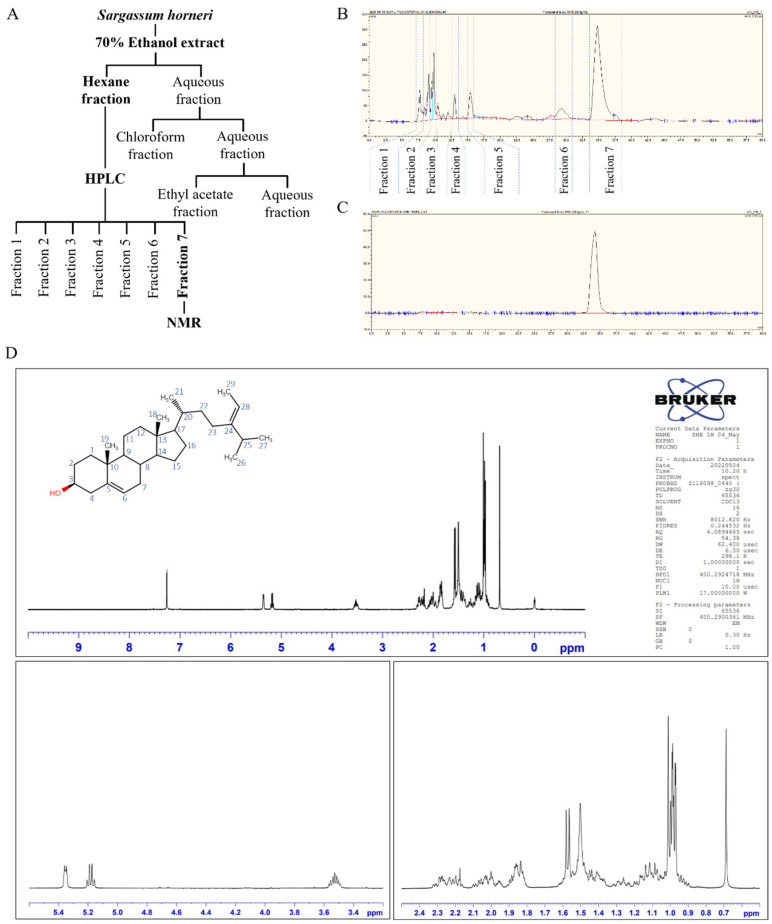
(**A**) Flow diagram representing the extraction and fractionation of *S. horneri* 70% ethanol extract. (**B**) HPLC chromatogram of hexane fraction. (**C**) HPLC chromatogram of prominent peak collected from hexane fraction and, (**D)** pure compound was characterized using ^1^H NMR analysis. All experiments were performed in triplicate (*n* = 3) to determine repeatability.

**Figure 2 antioxidants-11-01429-f002:**
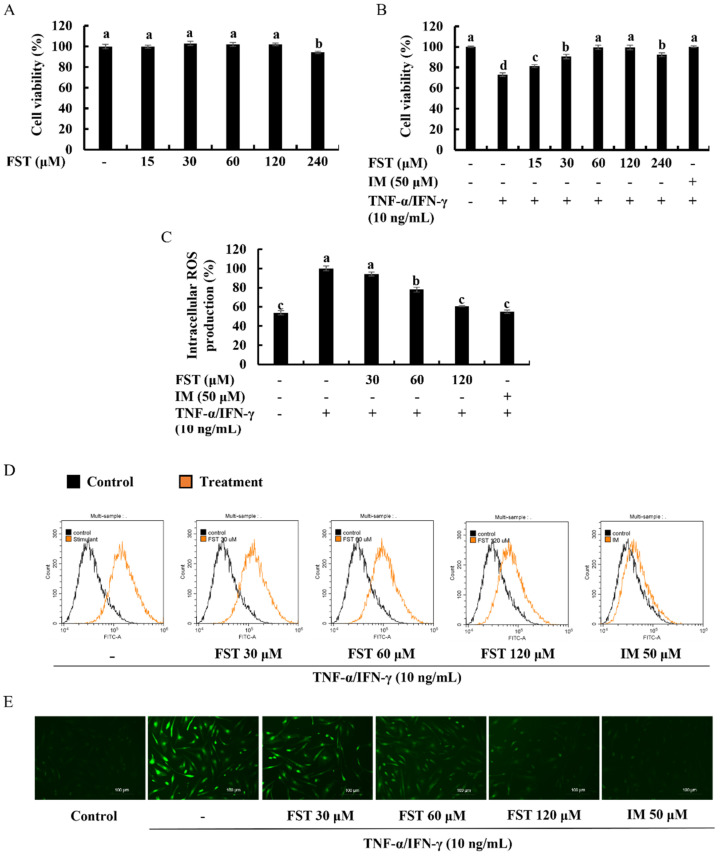
Cytoprotective effect of FST on human dermal fibroblasts (HDF). (**A**) Cytotoxicity, (**B**) cell viability (%), and analysis of intracellular ROS generation through (**C**) fluorometry, (**D**) flow cytometry analysis, and (**E**) fluorescence microscopy of FST-pre-treated TNF-α/IFN-γ-stimulated HDF cells. Indomethacin (IM, 50 μM) was used as a positive control. All experiments were performed in triplicate (*n* = 3) to determine the repeatability and lettered error bars were significantly different (*p* < 0.05).

**Figure 3 antioxidants-11-01429-f003:**
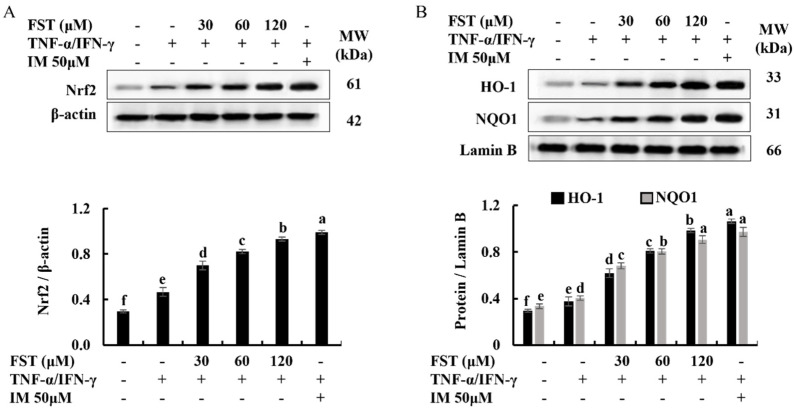
Dose-dependent effect of FST on Nrf2-mediated activation of HO-1 and NQO1 in TNF-α/IFN-γ-stimulated HDF cells. (**A**) Nrf2 expression, and (**B**) HO-1 and NQO1 expression. All experiments were performed in triplicate (*n* = 3) to determine the repeatability and lettered error bars were significantly different within the same molecule (*p* < 0.05).

**Figure 4 antioxidants-11-01429-f004:**
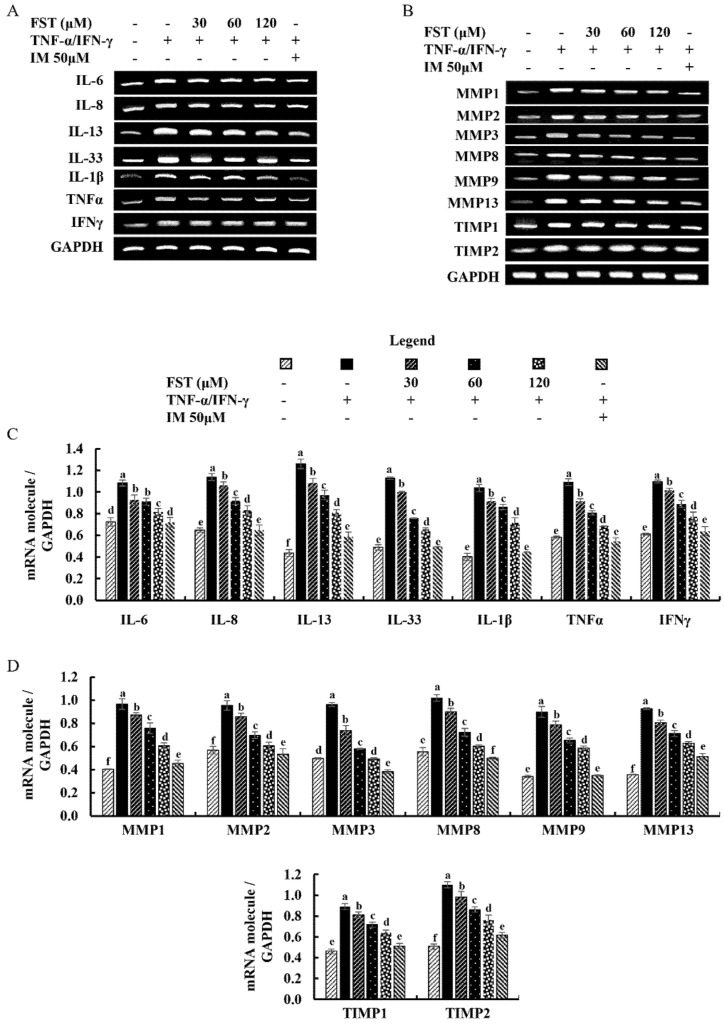
Inhibitory effect of FST on mRNA expression of inflammatory mediators, MMPs, and TIMP in TNF-α/IFN-γ-stimulated HDF cells. Effect of FST on (**A**) inflammatory cytokine expression, and (**B**) MMPs and TIMPs expression in TNF-α/IFN-γ-stimulated HDF cells evaluated by RT-PCR analysis. Densitometric analysis of (**C**) inflammatory cytokine expression, and (**D**) MMPs and TIMPs expression in TNF-α/IFN-γ-stimulated HDF cells. All experiments were performed in triplicate (*n* = 3) to determine the repeatability and lettered error bars were significantly different within the same molecule (*p* < 0.05).

**Figure 5 antioxidants-11-01429-f005:**
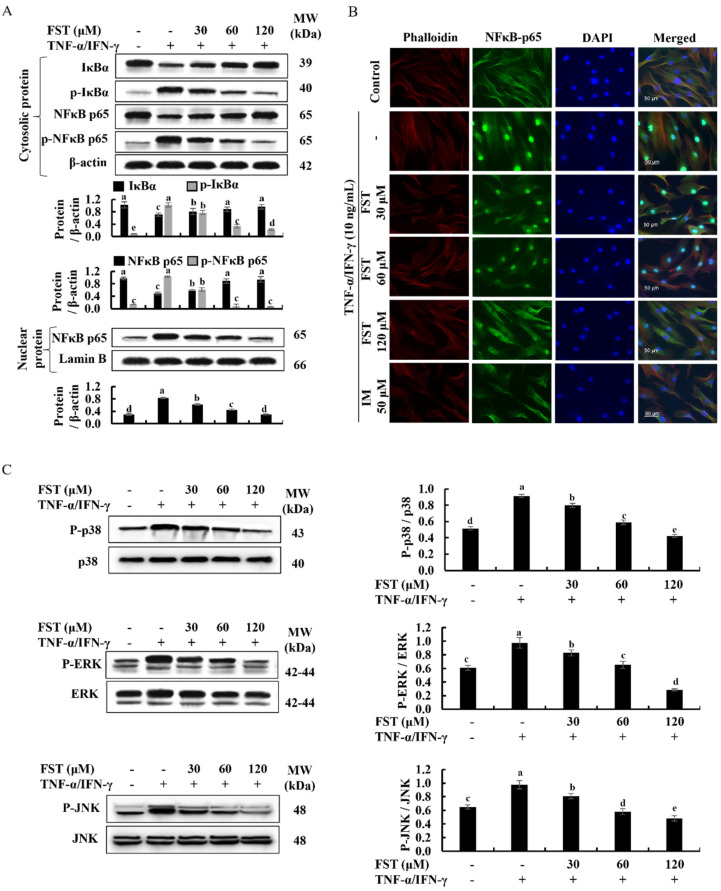
Protective effect of FST on NF-κB and MAPK inflammatory mediators in TNF-α/IFN-γ-stimulated HDF cells. Levels of molecular mediators were assessed by (**A**) western blot analysis for NF-κB, (**B**) evaluation of NF-κB p65 nuclear translocation by immunofluorescence analysis, and (**C**) western blot analysis of MAPK signaling. All experiments were performed in triplicate (*n* = 3) to determine the repeatability, and lettered error bars were significantly different (*p* < 0.05).

**Table 1 antioxidants-11-01429-t001:** Composition of SHE.

SHE	Composition %
Yield	8.12 ± 0.26
Carbohydrates	3.92 ± 0.19
Protein	1.09 ± 0.07
Total polyphenols	14.82 ± 0.68

Mean ± SEM (all experiments were performed in triplicate (*n* = 3) to determine the repeatability).

**Table 2 antioxidants-11-01429-t002:** Numerical values of spectra of the ^1^H-NMR values of FST.

No.	^1^H-NMR Value
1	1.81 (1H, m), 1.10 (1H, m)
2	1.52 (1H, m), 1.38 (1H, m)
3	3.51 (1H, m)
4	2.17 (1H, m), 2.07 (1H, m)
6	5.34 (1H, d)
7	1.89 (1H, m), 1.60 (1H, m)
8	1.43 (1H, m)
9	0.91 (1H, m)
11	1.50 (1H, m), 1.43 (1H, m)
12	1.99 (1H, m, 1.16 (1H, m)
14	1.01 (1H, m)
15	1.58 (1H, m), 1.07 (1H, m)
16	1.82 (1H, m), 1.26 (1H, m)
17	1.16 (1H, m)
18	0.69 (3H, s)
19	0.99 (3H, s)
20	1.40 (1H, m)
21	1.00 (3H, d)
22	1.41 (1H, m), 1.09 (1H, m)
23	2.03 (1H, m), 1.90 (1H, m)
25	2.20 (1H, m)
26	0.97 (3H, s)
27	0.97 (3H, s)
28	5.17 (1H, dd)
29	1.56 (3H, s)

All spectra were recorded in CDCL_3_ at 400 MHz.

## Data Availability

Data are included within the article.

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
