# Peer review of "Fucosterol Isolated from Dietary Brown Alga Sargassum horneri Protects TNF-α/IFN-γ-Stimulated Human Dermal Fibroblasts Via Regulating Nrf2/HO-1 and NF-κB/MAPK Pathways"

_antioxidants, 2022, doi:10.3390/antiox11081429_

Round 1
Reviewer 1 Report
In this original article, the authors have performed several experiments to demonstrate the protective effects of Fucosterol against TNF-α/IFN-γ-Stimulated Human Dermal Fibroblasts targeting Nrf2/HO-1 and NF-κB/MAPK Pathways. Although the overall approach to the research is proper scientifically sound, however I have a few concerns that needs to be addressed:
Comments and suggestions
1. English is poor….eg starting line in introduce “Science ancient times” …. Should it be since or science?
2. Line 47 and 49 have been cited with the same reference “4” …either reference should be removed from line 47 or should be cited alternatively.
3. As the authors have focused on Nrf2/HO-1 and NF-κB/MAPK Pathways, I will suggest describing pathways in brief.
4. How much total RNA was used to synthesize cDNA…I will suggest adding in the manuscript.
5. Resolution of figure 1. B and D are poor… must be improved
6. Molecular weight is missing in western blot images, must be added the specific molecular weight of all the proteins.
7. In western blot authors have used β-actin as housekeeping proteins, however, in gene expression GAPDH… did the author try to normalize the genes with β-actin?
8. Scale bar is missing in immunofluorescence images…. must be added.
Reviewer 2 Report
antioxidants-1839052
The paper entitled ‘Fucosterol Isolated from Dietary Brown Alga Sargassum horneri Protects TNF-α/IFN-γ-Stimulated Human Dermal Fibroblasts via Regulating Nrf2/HO-1 and NF-κB/MAPK Pathways.’ by the authors proposed that fucosterol purified from the edible brown alga may exhibit inflammatory effects by affecting known signaling pathways. The verification of the bioactivity of fucosteril seems to be based on the results of careful experiments. These results seem to be of a level of interest to related researchers. But I have some points should be considered by the authors before the publication process proceeds.
Major point
1. The degree of purification of fucosterol remains questionable. The purity of the substance to be analyzed must be at a level that guarantees the verification result of physiological activity. Purification by HPLC that made a single peak chromatogram was fine, but it is not possible to evaluate the purity based on this result alone. Since the authors conducted NMR analysis, they should show the numerical values of the spectral data and present a table comparing the numerical values of the existing spectra. Also, it should be clearly stated how many peaks are derived from impurities.
In addition, I will give you some minor points that I noticed.
2. line 137: What it ‘ before Ultimate?
3. line 202 and 216: What is “SHE”?
4. In lune 242, the abbreviation for human dermal fibroblasts is HDF, but he has already appeared in line 224. Check out other abbreviations as well. Also, if possible, please provide a list of abbreviations.
Round 2
Reviewer 1 Report
Authors have responded all my comments in scientific and positive way, and hence manuscript has improved significantly.
Reviewer 2 Report
The authors addredded the review's comments.